# Comparative Study of Patients with Periodontal and Parkinson’s Disease: Clinical and Salivary Aspects

**DOI:** 10.3390/biomedicines13122885

**Published:** 2025-11-26

**Authors:** Dragoș Nicolae Ciongaru, Silviu Mirel Piţuru, Stana Păunică, Marina Cristina Giurgiu, George Alexandru Denis Popescu, Anca Silvia Dumitriu

**Affiliations:** 1Department of Periodontology, Faculty of Dental Medicine, Carol Davila University of Medicine and Pharmacy, 050474 Bucharest, Romania; nicolae-dragos.ciongaru@drd.umfcd.ro (D.N.C.); stana.paunica@umfcd.ro (S.P.); anca.dumitriu@umfcd.ro (A.S.D.); 2Department of Professional Organization and Medical Legislation-Malpractice, Faculty of Dental Medicine, Carol Davila University of Medicine and Pharmacy, 050474 Bucharest, Romania; silviu.pituru@umfcd.ro; 3Department of General Practice I, Faculty of Medicine, Carol Davila University of Medicine and Pharmacy, 050474 Bucharest, Romania; george-denis.popescu@drd.umfcd.ro

**Keywords:** Parkinson’s disease, periodontal diagnosis, clinical indices, salivary aspects

## Abstract

**Introduction**: This study investigates the severity of periodontal disease in patients with Parkinson’s disease by comparing clinical and salivary aspects. **Materials and Methods**: A total of 31 patients were included: 15 with periodontal disease (control group) and 16 with periodontal disease and Parkinson’s disease (study group). Demographic data, periodontal parameters (plaque index, tartar index, bleeding index, probing depth, periodontal pocket index) and salivary parameters included viscosity, pH, and buffering capacity were analyzed. **Results:** Patients with Parkinson’s disease exhibited slightly lower mean values for plaque accumulation, bleeding on probing, and tartar index compared with the control group, though these differences were not statistically significant. In contrast, salivary parameters, particularly buffering capacity, showed statistically significant differences (*p* < 0.05) between the groups. **Conclusions:** Parkinson’s disease impacts periodontal health. Early intervention and integrated care strategies may help mitigate oral health deterioration in Parkinson’s patients.

## 1. Introduction

This study examines the hypothesis that periodontal disease is linked to significant alterations in clinical and salivary indices in patients with Parkinson’s disease compared to those without the condition. Periodontal disease (PD) and Parkinson’s disease are interconnected conditions with potentially shared pathogenic mechanisms. Periodontal disease, characterized by inflammation of the supporting structures of the teeth, has been associated with various systemic health conditions, including neurodegenerative disorders [1]. Parkinson’s disease, a progressive neurological condition, has also been associated with alterations in oral health, possibly due to motor impairments and systemic inflammatory responses [2]. Clinical and histopathological evidence supports the idea that periodontal inflammation may contribute to neuronal dysfunction and degeneration [3,4]. In addition, immunological perspectives highlight the role of the immune system in the complex interaction between these two conditions. Parkinson’s disease is the second most common neurodegenerative disorder. The condition predominantly affects dopaminergic neurons, leading to a gradual degradation of neural connectivity. This degeneration results in motor, cognitive, and psychiatric symptoms, with postural instability becoming increasingly apparent as the disease progresses [5,6]. Multiple mechanisms have been implicated in the pathogenesis of both periodontal and Parkinson’s disease. Although periodontal disease involves a diverse microbiota, Gram-negative bacteria play a significant pathogenic role [7]. These bacteria release toxins that contribute to local and systemic inflammation and stimulate the production of pro-inflammatory mediators [1]. The imbalance of these mediators forms the basis of the proposed association between periodontal disease and other systemic conditions, including heart disease, diabetes, low birth weight, and neurodegenerative diseases [8]. Periodontal disease poses significant health challenges across diverse populations. With its multifactorial etiology involving genetic, environmental, and behavioral factors, its prevalence is markedly elevated among certain high-risk groups, including those with systemic conditions [8,9]. Emerging research indicates that patients with Parkinson’s disease exhibit distinct oral health challenges arising from multiple pathways, including motor dysfunction, altered cognitive abilities, and changes in salivary flow and composition [10,11]. These factors impair daily oral hygiene practices and may trigger inflammatory cascades that further compromise periodontal health. Emerging evidence highlights the role of *Porphyromonas gingivalis* in linking periodontal disease to neurodegenerative disorders such as Parkinson’s disease. *P. gingivalis* produces potent virulence factors, including lipopolysaccharides (LPS) and gingipains, which can enter the systemic circulation and initiate inflammatory responses [12]. LPS from Gram-negative bacteria has been shown to induce microglial activation, oxidative stress, and apoptosis in neuronal tissues—processes that are central to the pathophysiology of Parkinson’s disease. Furthermore, experimental studies suggest that bacterial inflammation may promote the misfolding, aggregation, and propagation of α-synuclein—a pathological hallmark of Parkinson’s disease—within both the central and peripheral nervous systems [13]. Gingipain-mediated cleavage of host proteins and LPS-induced mitochondrial dysfunction may additionally exacerbate dopaminergic neuronal vulnerability. These mechanistic pathways support a biologically plausible link through which chronic periodontal inflammation could influence neurodegeneration, offering a potential explanation for the bidirectional association observed between periodontal disease and Parkinson’s disease. Understanding the dynamics of these interactions is essential for developing targeted interventions aimed at improving the quality of life of patients affected by these comorbid conditions. The hypothesis that these alterations may arise from both impaired oral hygiene and heightened systemic inflammation raises important questions regarding the underlying mechanisms. The rationale for the present study lies in elucidating the intersection between neurodegenerative and inflammatory mechanisms at the oral level. By investigating clinical periodontal parameters in relation to salivary properties such as buffering capacity and viscosity, this research aims to advance an integrated understanding of oral–systemic interactions in Parkinson’s disease. The principal aim of the study is to compare the clinical and salivary characteristics of patients with periodontal disease, with and without Parkinson’s disease, to identify specific oral alterations associated with the neurodegenerative condition. The specific objectives are (a) to assess and compare key periodontal indices, including plaque accumulation, bleeding on probing, probing depth, and periodontal pocket index, between the two groups; (b) to evaluate salivary parameters: pH, viscosity, and buffering capacity, and determine differences; (c) to identify potential correlations between clinical and salivary indices that may reflect systemic inflammatory influences in patients with Parkinson’s disease; and (d) to outline the clinical implications of these findings for preventive and therapeutic management strategies aimed at maintaining oral health and improving the quality of life of affected individuals.

In summary, as the growing body of literature increasingly emphasizes the interplay between oral health and systemic diseases such as Parkinson’s, it becomes imperative to deepen our understanding of this relationship. Exploring the pathways connecting these two facets of health not only informs clinical practice but also supports the development of approaches to improving well-being in individuals experiencing both Parkinson’s disease and periodontal disease.

## 2. Materials and Methods

### 2.1. Study Design

An observational, analytical, and cross-sectional approach was adopted to explore the relationship between periodontal status and Parkinson’s disease. The study design was not blinded, as patient interaction, clinical neurological features, and medical history made it impossible to conceal Parkinson’s disease status.

### 2.2. Participants

The current research was conducted at the Faculty of Dentistry, “Carol Davila” University of Medicine and Pharmacy in Bucharest, Romania. Patients were diagnosed and treated at the Periodontology Department of the Emergency University Hospital of Bucharest. Throughout the study period, a total of 450 patients visited the Department of Periodontology. Based on the criteria outlined in our study protocol, we initially identified 59 potential participants. Ultimately, 31 patients met the inclusion criteria and were selected for the study, comprising 15 patients with periodontal disease and 16 patients with both periodontal and Parkinson disease. The remaining 28 patients were excluded for various reasons: 5 due to acute form of periodontal diseases, 14 because of systemic conditions affecting periodontal treatment (such as diabetes or cardiovascular diseases) and 9 for lack of cooperation. All patients in the Parkinson’s disease group were receiving dopaminergic therapy consisting of levodopa–carbidopa, a combination of a dopamine precursor and a peripheral decarboxylase inhibitor.


*Inclusion Criteria*


Adults aged 18–80 years.Confirmed diagnosis of periodontal disease according to the 2018 EFP classification [14].Confirmed diagnosis of Parkinson’s disease established by a specialized neurologist.Adequate oral hygiene: patients or their caregivers.No systemic comorbidities.No antibiotic therapy within the last 12 months.No periodontal treatment (antimicrobial or surgical) within the last 12 months.Absence of implants or fixed prosthetic restorations.


*Exclusion Criteria*


Current or former smokers.History of systemic diseases other than Parkinson’s disease.Severe motor impairment preventing compliance with study procedures.Presence of acute periodontal conditions (e.g., abscesses).Use of monoamine oxidase inhibitors (MAOIs) or other medications contraindicated for the study.Pregnancy.Known allergy to antibiotics.

Two groups of patients were recruited: Control group: Patients diagnosed exclusively with periodontal disease (n = 15); Study group: Patients with both periodontal disease and Parkinson’s disease (n = 16).


*Ethical Considerations*


The study protocol was approved by the Scientific Research Ethics Commission of the “Carol Davila” University of Medicine and Pharmacy, Bucharest, Romania (Protocol number: 22334/05.09.2025), and was guided in accordance with the Declaration of Helsinki of 1975. All patient recruitment and data collection were conducted strictly after receiving approval from the Scientific Research Ethics Commission, and no data were collected prior to this date. Informed consent was obtained from all participating patients. Each patient demonstrated adequate cognitive understanding and motor ability to comprehend and personally sign the consent form.

### 2.3. Data Collection

#### 2.3.1. Demographic Data

Anamnesis provided information on patients’ age, sex, and duration since the diagnosis of Parkinson’s disease. The medical history confirmed that Parkinson’s disease had been diagnosed by a specialist neurologist.

#### 2.3.2. Clinical Indices

All examinations were performed by a single experienced periodontist. The following clinical parameters were assessed at baseline: number of teeth lost due to periodontal disease, periodontal disease staging and grading according to the periodontal classification [14], bleeding on probing/bleeding index, plaque index, tartar index, periodontal pocket index, and probing depth. In addition, the reason for presentation was recorded from the medical history.

All measurements were performed by the same examiner using a periodontal probe (North Carolina 15 mm probe) on six sites for each tooth. Bleeding on probing: Measures the presence or absence of bleeding when probing the gingival sulcus, indicating inflammation and potential periodontal disease activity [15]. Plaque Index: Assesses the visible accumulation of dental plaque on the tooth surfaces and serves as an indicator of oral hygiene and potential inflammatory response [8]. Tartar Index: Evaluates the degree of mineralized deposits (tartar) on teeth, which can contribute to gum inflammation and periodontal disease if not removed [8]. Probing Depth: Measured with a calibrated periodontal probe, indicates the depth of the periodontal pocket. Increased probing depth can signal inflammation and epithelial attachment loss. The probing depth is measured from the level of the gingival margin to the most apical point of the gingival sulcus or periodontal pocket [16]. Periodontal Pocket Index: Indicates the presence of periodontal pockets and helps assess the extent of periodontal disease and inflammation. Considering that a periodontal pocket is defined as having a probing depth greater than 3 mm, the periodontal pocket presence index is evaluated as the number of surfaces with pockets divided by the total number of dental surfaces, multiplied by 100 [8].

#### 2.3.3. Salivary Indices

Salivary indices were determined using the GC Saliva-Check Buffer Test, following the manufacturer’s instructions and recommendations. All examinations were performed by a single experienced periodontist.

Saliva collection and assessment were standardized to minimize variability. All samples were collected in the early afternoon, after participants had abstained from food, beverages, and oral hygiene procedures for at least two hours. All patients receiving dopaminergic therapy took their regular morning medication to ensure a stable pharmacological condition during collection.

##### pH

Saliva was collected in a sterile container. The Buffer Test kit uses pH indicator strips for determination. Unstimulated saliva was collected in the provided collection cup. A pH test strip was then taken from the kit and immersed in the saliva sample for 10 s to ensure proper reading. Upon removal, the strip was immediately compared with the color chart on the kit packaging, and the corresponding value was recorded as the resting salivary pH.

##### Viscosity

Salivary viscosity was assessed qualitatively using the GC Buffer test. The procedure involves collecting unstimulated saliva, observing its appearance and consistency. Salivary viscosity was evaluated visually using the descriptive scale provided in the GC Saliva-Check Buffer Test, classifying samples as normal (watery, clear) or high (sticky, frothy, or stringy) according to their macroscopic appearance and adherence to the collection surface. All evaluations were performed by the same calibrated examiner under consistent environmental conditions.

This method is subjective and may vary depending on the clinician’s interpretation.

##### Buffering Capacity

Buffering capacity was measured with the GC kit to assess acid-neutralizing ability (similar procedure as the pH test but measuring buffering capacity instead of direct pH). The kit assesses buffering capacity using a colorimetric method. Unstimulated saliva was collected in a clean container according to the kit instructions. A buffer test strip containing multiple test pads was then retrieved from the kit. Using the pipette provided, one drop of saliva was applied to each test pad, after which the strip was gently tilted to allow excess saliva to drain without wiping the pads. The strip was left undisturbed for approximately two minutes, as specified by the manufacturer. Subsequently, the color of each test pad was compared to the reference chart included in the kit, with each pad assigned a point value corresponding to its color.

### 2.4. Statistical Analysis

All data were analyzed using IBM SPSS Statistics 25 and illustrated with Microsoft Office Excel/Word 2024. Qualitative variables included sex, salivary viscosity, buffering capacity, reason for visit, and periodontal classification (stage and grade). These were expressed as counts or percentages and compared between groups using Fisher’s Exact Test; Z-tests with Bonferroni correction were applied for post hoc analysis of contingency tables. Quantitative variables included age, duration of Parkinson’s disease, number of missing teeth, plaque index, tartar index, bleeding index, probing depth, pocket presence index, and salivary pH. These were expressed as means with standard deviations or as medians with interquartile ranges. Normality of distributions was assessed with the Shapiro–Wilk Test. Quantitative independent variables with non-parametric distributions were compared between groups using the Mann–Whitney U Test, while those with normal distributions were tested using the Student’s *t*-Test, after verifying homogeneity of variances with Levene’s Test.

## 3. Results

### 3.1. Clinical Indices

The study included a total of 31 patients diagnosed with periodontal disease, of whom 16 (51.61%) also had Parkinson’s disease and 15 presented only periodontal pathology without neurological involvement. A comparative analysis was performed to assess potential clinical and salivary differences between the two groups, using statistical methods appropriate to the type and distribution of the variables (Table 1). The mean age of participants was 61.1 ± 11.06 years, reflecting a middle-aged to elderly population, typically at higher risk for both periodontal and neurodegenerative diseases. This difference indicates that the two groups were comparable in terms of age distribution. Sex distribution showed that 29% of all participants were male and 71% female. In the control group, 33.3% were male, compared to 25% in the Parkinson’s group. The difference was not statistically significant (*p* = 0.704), suggesting that gender did not introduce bias between the groups. In the Parkinson’s group, the mean disease duration was 7.81 ± 2.19 years, reflecting an intermediate stage of progression, which may influence both general and oral health. The median number of missing teeth was 5 (IQR 4–8), with no significant difference between groups (*p* = 0.281). In the control group, the median was 6 (IQR 5–8), compared to 5 (IQR 2.25–11.25) in the Parkinson’s group. The mean tartar index across all patients was 58.06 ± 27.78, with higher values in the control group (67.4 ± 25.03) compared to the Parkinson’s group (49.31 ± 28.11). Although the difference approached statistical relevance, it did not reach significance (*p* = 0.069). The plaque index was high in both groups, with a mean value of 67.42 ± 25.37 overall. The control group had a mean of 68.93 ± 24.66, and the Parkinson’s group 66 ± 26.74 (*p* = 0.754). Similarly, the bleeding index averaged 50.29 ± 22.96, with higher values in the control group (56.67 ± 22.83) than in the Parkinson’s group (44.31 ± 22.13). Median probing depth was 5 mm (IQR 4–6), with values of 6 mm in controls and 5 mm in Parkinson’s patients (*p* = 0.264). The pocket presence index had a median of 28 (IQR 12–52), slightly higher in the control group than in the Parkinson’s group.

The most common reason for presentation was gingival bleeding (58.1%), followed by tooth mobility (25.8%) and routine control (16.1%). Distribution was comparable between groups (Figure 1).

Regarding periodontal staging, most patients were classified as Stage III (61.3%), followed by Stage II (29%) and Stage IV (9.7%). In terms of grading, the vast majority of patients were Grade B (90.3%), while 9.7% were Grade C.

### 3.2. Salivary Indices

The analysis of salivary characteristics revealed relevant differences. Salivary viscosity was normal in 54.8% of patients and high in 45.2%. In the control group, 60% showed normal viscosity compared to 50% in the Parkinson’s group. Although the difference did not reach statistical significance (*p* = 0.722), the tendency toward higher viscosity in Parkinson’s patients suggests a potential alteration in salivary quality associated with the neurological condition (Figure 2).

Mean salivary pH was 6.88 ± 0.35, with nearly identical values in both groups (6.88 ± 0.32 in the control group and 6.88 ± 0.39 in the Parkinson’s group). No statistically significant difference was observed (*p* = 0.995), indicating that resting salivary pH was not influenced by Parkinson’s disease. In contrast, buffering capacity demonstrated significant intergroup variation (Figure 2). Overall, 54.8% of patients presented normal buffering capacity, 35.5% low, and 9.7% very low. Statistical analysis revealed significant differences (*p* = 0.035), confirmed by Bonferroni-corrected Z-tests. Low buffering capacity was strongly associated with Parkinson’s disease (56.3% vs. 13.3% in controls), whereas normal buffering capacity was more frequent in patients without Parkinson’s disease (73.4% vs. 37.5%). These findings indicate that salivary buffering ability, a key protective factor against oral disease, is compromised in patients with Parkinson’s disease.

## 4. Discussion

This study investigated periodontal and salivary parameters in patients with and without Parkinson’s disease, aiming to clarify how neurological impairment may influence periodontal inflammation and oral health. Although periodontal indices did not differ statistical significance between groups, a clear pattern emerged: patients with Parkinson’s disease displayed more profound inflammatory alterations, particularly reflected in their salivary profiles, which may predispose them to periodontal deterioration. The demographic characteristics of the two groups were comparable in terms of age and sex distribution, supporting the internal validity of the comparisons. Both groups consisted predominantly of middle-aged and older adults, a population naturally more susceptible to periodontal disease due to age-related immune dysregulation and cumulative inflammatory exposure. Periodontal indices such as plaque, bleeding on probing, probing depth, and periodontal pocket index did not show statistically significant differences between groups. However, when interpreting these data from a biological perspective, it is important to recognize that patients with Parkinson’s disease tended to demonstrate poorer clinical performance. For example, bleeding on probing, a direct marker of gingival inflammation, was elevated in many Parkinson’s patients, reflecting chronic inflammation and exacerbated host inflammatory response. Likewise, probing depths and periodontal pocket index also indicates periodontal tissue breakdown, and although the numerical differences were modest, the combination of altered salivary environment and systemic inflammation suggests that these lesions may be more difficult to control and more prone to progression in Parkinson’s disease.

It is also noteworthy that the entire study population presented advanced periodontal involvement, with most patients classified as Stage III, Grade B periodontitis [14]. This overall disease severity may have masked statistical differences between the groups. Nonetheless, within this context of generalized periodontal, patients with Parkinson’s disease appear more vulnerable due to the systemic and local inflammatory environment associated with their condition.

The most significant findings of the study are related to salivary buffering capacity. Patients with Parkinson’s disease showed a markedly higher prevalence of low buffering capacity (56.3% vs. 13.3% in controls). Buffering is essential for neutralizing bacterial acids and protecting oral tissues from chronic inflammation [17,18,19]. Reduced buffering shifts the oral microenvironment toward acidity, promoting pathogenic bacterial growth, collagen degradation, and persistent gingival inflammation [20,21,22].

Salivary viscosity was also altered in the Parkinson’s group. Low viscosity compromises the natural cleansing function of saliva and increases bacteria adherence to gingival surfaces, thereby sustaining chronic inflammatory response [22]. Although differences in viscosity and pH were not significant, the clinical tendency is evident: patients with Parkinson’s disease experience a more hostile oral environment in which saliva is less capable of performing its protective, anti-inflammatory roles. These pharmacologically mediated alterations create an inflammatory cascade in the oral cavity: reduced saliva flow increases plaque retention, promotes bacterial dysbiosis, and lead to gingival inflammation. Additionally, while the salivary pH did not differ significantly between groups, buffering capacity was markedly reduced in Parkinson’s patients. This apparent discrepancy suggests that qualitative or compositional changes in saliva—such as altered protein or bicarbonate content—may compromise its neutralizing potential despite a similar resting pH [22,23]. These findings highlight the importance of evaluating not only salivary pH but also its functional parameters when assessing oral inflammatory risk in Parkinson’s disease.

Taken together, these findings align with the hypothesis that Parkinson’s disease, beyond its well-known effects on motor control, significantly influences inflammatory regulation within the oral cavity.

However, it is important to acknowledge that the assessment relied exclusively on the GC Saliva-Check Buffer Test, a qualitative chairside method. Although this test provides practical and rapid clinical information, it lacks the analytical precision of laboratory-based titration methods and may be influenced by examiner subjectivity and environmental factors. Therefore, while the results strongly suggest impaired buffering capacity in the Parkinson’s group, these findings should be interpreted with caution and validated in future studies using standardized quantitative techniques. Finally, although the GC Saliva-Check Buffer Test provided practical and standardized chairside assessment of salivary characteristics, its semi-quantitative nature represents a methodological limitation. This test has demonstrated good reproducibility and acceptable correlation with laboratory titration methods according to previous studies [24,25]. Consequently, while the observed reduction in buffering capacity remains a key and statistically significant finding, it should be interpreted cautiously. Future investigations should validate these results using both chairside and laboratory-based methods to ensure a more comprehensive evaluation of salivary function in Parkinson’s disease. It should also be noted that all salivary parameters and indices were determined using unstimulated saliva collected within the GC Saliva-Check Buffer test. This approach provides valuable information regarding baseline salivary characteristics but may not fully capture the dynamic variations observed during stimulated secretion. Future studies could therefore benefit from including both unstimulated and stimulated saliva samples to obtain more detailed and functionally relevant data on salivary composition and buffering performance.

Patients with systemic comorbidities and smoking history were excluded to reduce potential confounding factors known to influence both periodontal inflammation and salivary composition. These exclusions were intended to isolate the specific contribution of Parkinson’s disease to oral health alterations. However, this methodological choice inevitably restricts the external validity of the findings, as comorbid conditions and smoking are highly prevalent in the elderly Parkinsonian population. Because all patients with systemic comorbidities were excluded, the study population represents a highly selected group, which severely limits the generalizability of the findings. Therefore, the results should be interpreted as applying only to a restricted subset of patients with periodontal disease and Parkinson’s disease who do not present additional systemic conditions. Future studies should therefore include stratified analyses that reflect the broader clinical reality of these patients.

As noted, all patients with Parkinson’s disease in this study were under dopamine-based pharmacological therapy. Levodopa and related agents, while essential for motor control, frequently induce xerostomia, modify salivary viscosity, and reduce buffering capacity. The potential impact of dopaminergic therapy on salivary and periodontal parameters warrants further investigation. The exact duration of antiparkinsonian medication use is challenging to determine, as it generally mirrors the time elapsed since diagnosis (mean: 7.81 ± 2.19 years, as shown in Table 1). All patients in the present study received levodopa–carbidopa, commercially known as Isicom or derivatives from the same class of medications, with dosages individualized according to disease severity—ranging from 1/2 tablet (125 mg/12.5 mg) daily in early stages to 2 tablets (250 mg/25 mg) daily in advanced stages, under the supervision of a neurologist. The therapy was typically administered twice daily, in the morning and evening, following the neurologist’s recommendations. Despite its central role in Parkinson’s disease management, the influence of dopaminergic medication on salivary secretion and oral homeostasis remains poorly understood. To date, there is no clear or conclusive evidence demonstrating that levodopa–carbidopa exert a direct and consistent effect on salivary flow, buffering capacity, viscosity, or overall salivary composition. Existing studies are limited, often heterogeneous, and report conflicting results, suggesting that any potential salivary alterations may depend on a complex interplay of disease severity, autonomic dysfunction, hydration status, medication timing, and individual patient variability rather than on medication alone. The sample size reflects the total number of eligible patients meeting strict inclusion criteria within the recruitment timeframe, ensuring methodological rigor despite the limited cohort. Given these uncertainties, future research should prioritize well-designed, adequately powered studies that investigate the relationship between dopaminergic drug type, dosage, timing of administration, polypharmacy, and long-term treatment duration and the observed variations in salivary properties and periodontal status in patients with Parkinson’s disease. Such studies are essential to clarify whether salivary dysfunction arises primarily from the neurodegenerative process itself, the pharmacological therapy, or their combined systemic effects.

Thus, the interplay between systemic inflammation, local inflammatory imbalance, and salivary dysfunction explains why Parkinson’s patients are much more affected than controls, even when periodontal indices appear similar. The underlying inflammatory burden makes their periodontal tissues less resilient and more prone to progressive destruction.

These findings underline that conventional periodontal indices alone may underestimate the vulnerability of Parkinson’s patients. While bleeding on probing or probing depth may appear comparable, the reduced salivary buffering and slight increased viscosity in the Parkinson’s group reveal a heightened inflammatory potential that can accelerate disease progression. In practice, this means that patients with Parkinson’s disease require earlier and more intensive preventive interventions to counteract this hidden inflammatory risk. Integrated care should include not only mechanical plaque control but also therapeutic strategies to modulate the oral inflammatory environment: salivary stimulants, saliva substitutes, remineralizing and anti-inflammatory agents, and close monitoring of periodontal tissues. Collaboration between dentists, neurologists, and caregivers is essential, since pharmacological regimens and motor impairments directly impact oral inflammation and hygiene ability.

Several methodological aspects should be acknowledged when interpreting the present findings. First, the total sample size was limited and determined by the number of eligible patients attending the university hospital. As a cross-sectional analysis, this study did not include an a priori power calculation. Consequently, the sample may be underpowered to detect small but clinically relevant differences between groups. Nevertheless, this approach is consistent with previous exploratory investigations on oral and salivary alterations in Parkinson’s disease [26,27,28]. The relatively small sample size of this study limited the statistical power to detect subtle intergroup variations. Moreover, the advanced periodontal disease observed in both groups may have obscured clearer differences. Stratification according to medication type and disease severity will also help disentangle the effects of pharmacological treatment from those of the underlying neurodegenerative condition. Due to the limited sample size, a multivariate logistic regression model assessing the independent association between Parkinson’s disease and buffering capacity could not be performed. Future studies with larger cohorts should include multivariate analyses adjusting for age, sex, tooth number, plaque accumulation, medication type and dosage, and disease duration to more accurately determine the independent contribution of Parkinson’s disease to impaired salivary function. The number of participants was determined by the total eligible patients attending the Periodontology Department of the University Hospital during the study period. Consequently, the sample reflects a convenience-based clinical population rather than a statistically powered cohort representative of the general population. Future research should include formal sample size estimation based on expected effect size, statistical power, and confidence level to ensure adequate sensitivity for detecting intergroup differences.

Nevertheless, the present findings strongly suggest that Parkinson’s patients experience more pronounced inflammatory alterations, particularly in salivary function, which place them at greater risk for periodontal disease progression.

## 5. Conclusions

In conclusion, patients with Parkinson’s disease demonstrated significantly lower salivary buffering capacity compared to controls, despite showing no significant differences in conventional periodontal indices. This finding suggests that salivary impairment, rather than overt clinical periodontal parameters, may represent the earliest and most sensitive oral alteration in Parkinson’s disease. Recognizing and addressing this deficit is essential for preventing periodontal disease progression and maintaining oral health in this vulnerable population.

## Figures and Tables

**Figure 1 biomedicines-13-02885-f001:**
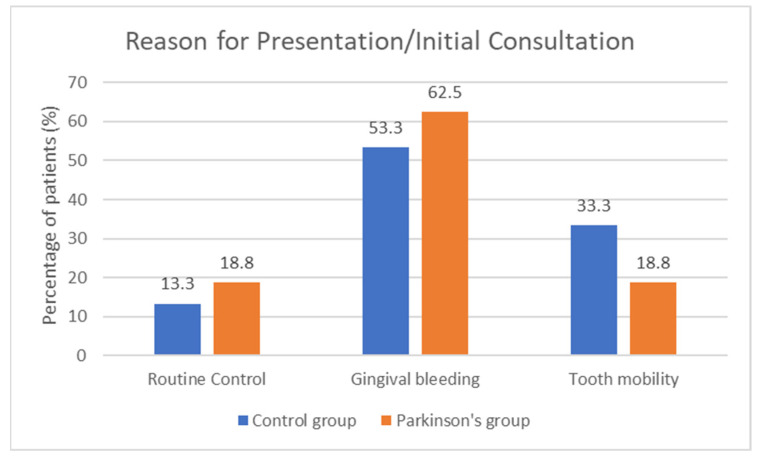
Distribution of reasons for presentation at the initial consultation in the control group and the group of patients with Parkinson’s disease. The most frequent reason was gingival bleeding (53.3% in the non-Parkinson’s group and 62.5% in the Parkinson’s group), followed by tooth mobility (33.3% vs. 18.8%) and routine control (13.3% vs. 18.8%). No statistically significant differences were observed between groups (*p* = 0.694). Values are expressed as percentages of the total number of patients.

**Figure 2 biomedicines-13-02885-f002:**
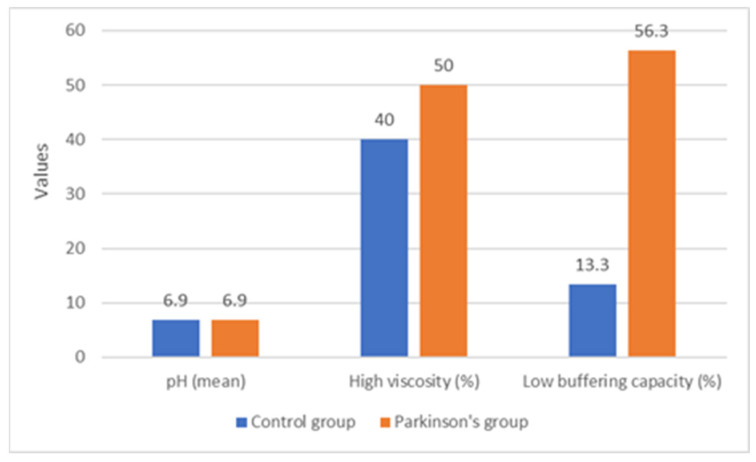
The comparison of salivary parameters between patients with and without Parkinson’s disease. The mean salivary pH values were nearly identical between the groups (6.88 ± 0.32 for the control group and 6.88 ± 0.39 for the Parkinson’s group; *p* = 0.995). The percentage of patients with increased salivary viscosity was 40.0% in the control group and 50.0% in the Parkinson’s group (*p* = 0.722). In contrast, reduced buffering capacity was significantly more frequent in patients with Parkinson’s disease (56.3%) compared to those without Parkinson’s (13.3%), representing a statistically significant difference (*p* = 0.035). Data are presented as means ± SD for pH and as percentages for the other variables.

**Table 1 biomedicines-13-02885-t001:** Comparison of analyzed characteristics according to studied groups.

Demographic Parameter	Total (N = 31)	Without Parkinson (N = 15)	With Parkinson (N = 16)	*p*
Age (Mean ± SD)	61.1 ± 11.06	63.33 ± 12.56	59 ± 9.36	0.283 *
Sex (Male) (Nr., %)	9 (29%)	5 (33.3%)	4 (25%)	0.704 **
Duration of Parkinson’s disease, years (Mean ± SD)	-	-	7.81 ± 2.19	-
**Salivary Parameters**	**Total**	**Without Parkinson**	**With Parkinson**	** *p* **
Viscosity (Nr., %)				
Normal	17 (54.8%)	9 (60%)	8 (50%)	0.722 **
High	14 (45.2%)	6 (40%)	8 (50%)	
pH (Mean ± SD)	6.88 ± 0.35	6.88 ± 0.32	6.88 ± 0.39	0.995 *
Buffering capacity (Nr., %)				
Very low	3 (9.7%)	2 (13.3%)	1 (6.2%)	0.035 **
Low	11 (35.5%)	2 (13.3%)	9 (56.3%)	
Normal	17 (54.8%)	11 (73.4%)	6 (37.5%)	
**Periodontal Parameter**	**Total**	**Without Parkinson**	**With Parkinson**	** *p* **
Missing Teeth (Median (IQR))	5 (4–8)	6 (5–8)	5 (2.25–11.25)	0.281 ***
Tartar Index (Mean ± SD)	58.06 ± 27.78	67.4 ± 25.03	49.31 ± 28.11	0.069 *
Plaque Index (Mean ± SD)	67.42 ± 25.37	68.93 ± 24.66	66 ± 26.74	0.754 *
Bleeding Index (Mean ± SD)	50.29 ± 22.96	56.67 ± 22.83	44.31 ± 22.13	0.137 *
Probing Depth (Median (IQR))	5 (4–6)	6 (5–6)	5 (4–6)	0.264 ***
Pocket Presence Index (Median (IQR))	28 (12–52)	32 (21–56)	22 (10.25–50.75)	0.401 ***
Reason for Visit (Nr., %)				
Control	5 (16.1%)	2 (13.3%)	3 (18.8%)	0.694 **
Gingival bleeding	18 (58.1%)	8 (53.3%)	10 (62.5%)	
Tooth mobility	8 (25.8%)	5 (33.3%)	3 (18.8%)	
Periodontitis Stage (Nr., %)				
Stage II	9 (29%)	3 (20%)	6 (37.5%)	0.479 **
Stage III	19 (61.3%)	11 (73.3%)	8 (50%)	
Stage IV	3 (9.7%)	1 (6.7%)	2 (12.5%)	
Periodontitis Grade (Nr., %)				
Grade B	28 (90.3%)	14 (93.3%)	14 (87.5%)	1.000 **
Grade C	3 (9.7%)	1 (6.7%)	2 (12.5%)	

* Student *t*-Test, ** Fisher’s Exact Test, *** Mann–Whitney U Test.

## Data Availability

Data are contained within the article.

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
