# Peer review of "Comparative Study of Patients with Periodontal and Parkinson’s Disease: Clinical and Salivary Aspects"

_biomedicines, 2025, doi:10.3390/biomedicines13122885_

Round 1

Reviewer 1 Report

Comments and Suggestions for Authors

Dear Authors,

Thank you for submitting your manuscript entitled “Comparative Study of Patients with Periodontal and Parkinson’s Disease: Clinical and Salivary aspects”. The topic is clinically important, but the paper in its current form has substantial methodological and reporting issue the prevent it from being suitable for publication. Below, I provide detailed comments to improve the clarity of the manuscript.

  • There is a critical inconsistency between the study period (reported as November 2022 – July 2025) and the ethical approval date (protocol no. 22334/05.09.2025). This would indicate that the study was completed before obtaining ethical approval. So, you must clarify the correct approval date and provide documentation, explicitly state whether the approval was retrospective and justify this, describe in detail how informed consent was obtained, especially for patients with possible cognitive impairment.
  • The total sample is underpowered for detecting clinically relevant differences. No power calculation or justification is provided. Please include an a priori power analysis or justify the sample size based on effect sizes from similar study.
  • The exclusion of participants with systemic comorbidities or smoking history strongly limits generalizability, particularly in an older Parkinson disease population. Please, justify these exclusions.
  • The GC Saliva-Check Buffer test is acceptable, but validation details are lacking. Please, provide information on the test’s reliability, sensitivity, and agreement with standard laboratory titration methods.
  • The saliva viscosity assessment appears subjective (“watery/sticky”). Please include a more detailed description of the scale used and any intra-examine reliability measures (e.g., ICC, kappa).
  • Clarify sample collection conditions (time of day, fasting state, hydration control, medication timing).
  • You state that “all PD patients were under dopaminergic therapy”, but critical details are missing: drug classes, dosages, treatment duration and use of xerostomia-inducing agents. These are essential, as saliva flow and oral hygiene are heavily drug- and severity- dependent.
  • A multivariate analysis should be performed to test the independent association between Parkinson’s disease and buffering capacity, adjusting for age, sex, tooth number, plaque index, medication use, and disease duration.
  • All examinations were performed by a single examiner. Please, indicate whether the examiner was blinded to the Parkinson’s diagnosis and provide calibration data or reliability metrics.
  • Ensure Figures 1-2 are high resolution, with complete captions, clear legends, and readable axis labels.
  • Table 1 should not be included in Materials and Methods but in Results.

Upon resubmission with these clarifications and analyses, I would be happy to reassess the revised version.

Kind regards.

Author Response

Dear Reviewer,

               Thank you for your observations. I have added the modifications and explanations based on your recommendations. Please read the revised version.

  • There is a critical inconsistency between the study period (reported as November 2022 – July 2025) and the ethical approval date (protocol no. 22334/05.09.2025). This would indicate that the study was completed before obtaining ethical approval. So, you must clarify the correct approval date and provide documentation, explicitly state whether the approval was retrospective and justify this, describe in detail how informed consent was obtained, especially for patients with possible cognitive impairment.

Response: We would like to clarify that the period initially reported (November 2022 – July 2025) referred exclusively to the general timeframe in which patients attended the Periodontology Department of the University Hospital and during which the study concept was developed and its necessity established. No patient recruitment, clinical examination, or data collection was performed prior to ethical approval. The study protocol was formally reviewed and approved by the Scientific Research Ethics Commission of the “Carol Davila” University of Medicine and Pharmacy, Bucharest, Romania (approval no. 22334/05.09.2025). All clinical procedures and data collection were conducted strictly after obtaining ethical approval, in full accordance with institutional and international ethical standards, including the Declaration of Helsinki (1975), as revised). All participants were thoroughly informed about the study objectives and procedures, both verbally and in writing, and each provided written informed consent prior to inclusion. Importantly, all individuals enrolled demonstrated full cognitive understanding and sufficient motor capacity to comprehend the study information and sign consent documents independently; therefore, no caregiver participation was required in the consent process. To prevent any further misunderstanding, the study period has been corrected and clarified in the revised manuscript, and the corresponding details regarding ethical approval and informed consent have been added to the Ethical Considerations section. The official ethical approval document and the signed informed consent forms were provided to the editorial office during submission to ensure complete transparency and compliance with ethical research standards.

  • The total sample is underpowered for detecting clinically relevant differences. No power calculation or justification is provided. Please include an a priori power analysis or justify the sample size based on effect sizes from similar study.

Response: Thank you for the observation. I have added relevant aspects related to these recommendations in the Discussion section. Please read the revised version.

  • The exclusion of participants with systemic comorbidities or smoking history strongly limits generalizability, particularly in an older Parkinson disease population. Please, justify these exclusions.

Response: I have added relevant aspects related to this observation in the Discussion section.

  • The GC Saliva-Check Buffer test is acceptable, but validation details are lacking. Please, provide information on the test’s reliability, sensitivity, and agreement with standard laboratory titration methods.

Response: I have added relevant aspects related to this observation in the Discussion section.

  • The saliva viscosity assessment appears subjective (“watery/sticky”). Please include a more detailed description of the scale used and any intra-examine reliability measures (e.g., ICC, kappa). Clarify sample collection conditions (time of day, fasting state, hydration control, medication timing).

Response: We thank the reviewer for this valuable observation. Additional methodological clarifications have been added in the Materials and Methods section to enhance reproducibility and transparency. Salivary viscosity was assessed using the visual descriptive scale included in the GC Saliva-Check Buffer Test, which categorizes saliva as normal (watery, clear) or high (sticky, frothy, or stringy), based on appearance and adherence to the container walls. Although qualitative in nature, the same experienced examiner performed all evaluations under standardized lighting and environmental conditions. Sample collection conditions have also been specified. These clarifications and methodological details have been incorporated into the revised manuscript.

  • You state that “all PD patients were under dopaminergic therapy”, but critical details are missing: drug classes, dosages, treatment duration and use of xerostomia-inducing agents. These are essential, as saliva flow and oral hygiene are heavily drug- and severity- dependent.

Response: We appreciate the reviewer’s insightful comment and have now added detailed information regarding dopaminergic therapy in the Materials and Methods and Discussion sections. All patients diagnosed with Parkinson’s disease were receiving treatment with levodopa–carbidopa combinations (commercially known as Isicom or pharmacologically equivalent formulations), which belong to the class of dopamine precursors and decarboxylase inhibitors. The treatment duration corresponded to the time since diagnosis, as reported in Table 1 (mean duration: 7.81 ± 2.19 years). The prescribed dosage of Isicom ranged from ½ tablet (125 mg/12.5 mg) daily in patients with mild-stage disease to 2 tablets (250 mg/25 mg each) per day in advanced stages, according to individual clinical requirements and neurologist supervision. We agree that future studies should explore the relationship between dopaminergic drug type, dosage, treatment duration, and their impact on salivary and periodontal parameters, as this may provide valuable insight into the pathophysiological mechanisms linking pharmacotherapy and oral health in Parkinson’s disease.

  • A multivariate analysis should be performed to test the independent association between Parkinson’s disease and buffering capacity, adjusting for age, sex, tooth number, plaque index, medication use, and disease duration.

Response: Thank you for the suggestion. Due to the small sample size, a multivariate analysis including multiple covariates would produce unstable and unreliable estimates. This limitation has been acknowledged in the revised Discussion, and we note that future adequately powered studies should incorporate multivariate modelling. Although a multivariate analysis could not be conducted in a statistically valid manner, we have explicitly acknowledged this limitation in the manuscript and clarified that future studies with larger, adequately powered samples are needed to explore the adjusted relationship between Parkinson’s disease and buffering capacity. We have added a sentence in the Discussion to highlight this aspect and align with your recommendation.

  • All examinations were performed by a single examiner. Please, indicate whether the examiner was blinded to the Parkinson’s diagnosis and provide calibration data or reliability metrics.

Response: The examinations were conducted by a single experienced periodontist; however, the study design was not blinded. The examiner was aware of each participant’s Parkinson’s disease status, as clinical neurological confirmation and medical history were part of the standard hospital records and could not be concealed during patient interaction. This type of observational, hospital-based clinical study does not allow for a blind or double-blind design, given that patient communication, motor symptoms, and medication disclosures make blinding impracticable. . We have added a sentence to highlight this aspect and align with your recommendation.

  • Ensure Figures 1-2 are high resolution, with complete captions, clear legends, and readable axis labels.

Response: Thank you for the observation. I have made the modification.

  • Table 1 should not be included in Materials and Methods but in Results.

Response: Thank you for the observation. I have made the modification.

Reviewer 2 Report

Comments and Suggestions for Authors

Dear authors,

Thank you for submitting your manuscript.  The manuscript addresses a clinically relevant topic, but I would like to point out some major concerns.

  • English Language Quality 

    The English language is inadequate and requires comprehensive professional editing. Errors are present throughout the entire manuscript.

  • Materials and methods

The study's most significant finding is the compromised salivary buffering capacity.    However, the reliance on the GC Saliva-Check Buffer Test, a qualitative/semi-quantitative chairside method, is a major methodological weakness. 

  • Results
The Abstract states that the study group (patients with Parkinson’s Disease, PD) showed "slightly increased values for clinical indices", including plaque accumulation, bleeding on probing, and probing depth, when compared with the control group. However, the data presented in Table 1 and detailed in the Results section clearly indicate the opposite: the control group (patients without PD) had numerically higher mean values for key clinical indices than the study group. 

The non-significant differences in Plaque index and Bleeding index are counter-intuitive given the known oral hygiene challenges in PD patients. The finding of a lower mean tartar index in the PD group is also surprising.

I find this manuscript currently undermined by significant methodological errors and not suitable for publication in this journal. 

Author Response

Dear Reviewer,

Thank you for your observations. I have added the modifications and explanations based on your recommendations. Please read the revised version.

  • English Language Quality . The English language is inadequate and requires comprehensive professional editing. Errors are present throughout the entire manuscript.

Response: I have reviewed the entire manuscript and made the revisions according to the recommendations.

  • Materials and methods. The study's most significant finding is the compromised salivary buffering capacity. However, the reliance on the GC Saliva-Check Buffer Test, a qualitative/semi-quantitative chairside method, is a major methodological weakness. 

Response: Thank you for the observation. I have added details regarding these aspects to the Discussion and Limitations sections.

  • Results. The Abstract states that the study group (patients with Parkinson’s Disease, PD) showed "slightly increased values for clinical indices", including plaque accumulation, bleeding on probing, and probing depth, when compared with the control group. However, the data presented in Table 1 and detailed in the Results section clearly indicate the opposite: the control group (patients without PD) had numerically higher mean values for key clinical indices than the study group. The non-significant differences in Plaque index and Bleeding index are counter-intuitive given the known oral hygiene challenges in PD patients. The finding of a lower mean tartar index in the PD group is also surprising.

Response: We have corrected this statement throughout the manuscript to reflect the findings

Reviewer 3 Report

Comments and Suggestions for Authors

The reviewer appreciates the effort of the author; however, after careful review, the reviewer has the following comments on the manuscript

Abstract:

Please add quantitative data with statistical significance in the results section.

Please rephrase the statement “The study group showed slightly increased values for clinical indices, including plaque accumulation, bleeding on probing, and probing depth, compared with the control group.  

Remove discussion from the abstract

Introduction:

The introduction section needs to be expanded by adding text stating an overview of previously published articles on this topic, research gap/ problem statement and rationale of the current study.

Add the aims and specific objectives of this study.

Please avoid adding multiple citations for a single statement that is not directly relevant to the statement. The citations should be specific and justify/ validate the scientific fact.

Methodology:

The author has expanded the patient flow of the university hospital without a proper calculation of the sample size. This kind of sample size is acceptable for a cohort study model. However, the current study title reflects a generalized population (cross-sectional study), In this type of study, the author must calculate the minimum number of samples based on effect size, statistical power, and confidence level. The reviewer highly recommends using G*Power software to calculate the correct sample size required for this research.  

Results:

Please add SD and statistical significance in Figures 1 and 2.

Discussion:

In the discussion section, the author stated several scientific facts without proper citations. On the other hand, there are multiple citations for a single statement that are not directly related to the point. The author should carefully check each reference cited in the manuscript to ensure its accuracy and relevance.

Author Response

Dear Reviewer,

               Thank you for your observations. I have added the modifications and explanations based on your recommendations. Please read the revised version.

  • Abstract: Please add quantitative data with statistical significance in the results section. Please rephrase the statement “The study group showed slightly increased values for clinical indices, including plaque accumulation, bleeding on probing, and probing depth, compared with the control group. Remove discussion from the abstract

Response: Thank you for the observation. I have made the modification.

  • Introduction:The introduction section needs to be expanded by adding text stating an overview of previously published articles on this topic, research gap/ problem statement and rationale of the current study. Add the aims and specific objectives of this study. Please avoid adding multiple citations for a single statement that is not directly relevant to the statement. The citations should be specific and justify/ validate the scientific fact.

Response: Thank you for the observation. I have made the modification.

  • Methodology:The author has expanded the patient flow of the university hospital without a proper calculation of the sample size. This kind of sample size is acceptable for a cohort study model. However, the current study title reflects a generalized population (cross-sectional study), In this type of study, the author must calculate the minimum number of samples based on effect size, statistical power, and confidence level. The reviewer highly recommends using G*Power software to calculate the correct sample size required for this research.  

Response: I have added relevant aspects related to this observation in the Discussion section.

  • Results:Please add SD and statistical significance in Figures 1 and 2.

Response: Thank you for the observation. I have made the modification.

  • Discussion: In the discussion section, the author stated several scientific facts without proper citations. On the other hand, there are multiple citations for a single statement that are not directly related to the point. The author should carefully check each reference cited in the manuscript to ensure its accuracy and relevance.

Response: Thank you for the observation. I have made the modification.

Round 2

Reviewer 1 Report

Comments and Suggestions for Authors

Dear​‍​‌‍​‍‌ authors,

Thank you for your thorough and considerate revision of the manuscript. The scientific quality and readability have been largely improved. Nevertheless, some issues still need to be addressed before the manuscript can be accepted. I kindly ask you to consider the following:

  • Ethical approval timing is still unclear. The text still does not make explicit that all patients were recruited and data collected after the approval of the Ethics Committee (05.09.2025). Please, add a clear sentence in the Ethical Considerations section stating that no data were collected before the approval.
  • Sample size justification is still inadequate. Please, add 1-2 sentences referring to effect sizes in similar studies or a post-hoc consideration about the feasibility of the sample size.
  • Exclusion criteria still limit the validity of the study. Please, clearly state that the exclusion of all patients with systemic comorbidities severely limits generalizability, and the results only apply to a very selected population.
  • The Results section still includes two almost identical paragraphs both describing Table 1. Please, merge them into a single, short description.
  • Language and style still need some revisions: some grammatical errors, repeated expressions, and long sentences are still a bit impeding. A final language revision is required to improve clarity and scientific readability.

Once these issues are solved, the manuscript will be ready for publication. Thank you for your effort in improving the manuscript, and I look forward to receiving the revised ​‍​‌‍​‍‌version.

Author Response

Dear Reviewer,

               Thank you for your observations. I have added the modifications and explanations based on your recommendations. Please read the revised version.

Ethical approval timing is still unclear. The text still does not make explicit that all patients were recruited and data collected after the approval of the Ethics Committee (05.09.2025). Please, add a clear sentence in the Ethical Considerations section stating that no data were collected before the approval.

Response: Thank you for the observation. I have added this aspect in the revised version.

Sample size justification is still inadequate. Please, add 1-2 sentences referring to effect sizes in similar studies or a post-hoc consideration about the feasibility of the sample size.

Response: Thank you for the observation. I have added this clarification to the Discussion section: “The sample size reflects the total number of eligible patients meeting strict inclusion criteria within the recruitment timeframe, ensuring methodological rigor despite the limited cohort.”

Exclusion criteria still limit the validity of the study. Please, clearly state that the exclusion of all patients with systemic comorbidities severely limits generalizability, and the results only apply to a very selected population.

Response: Thank you for this important observation. We agree that the exclusion of all patients with systemic comorbidities represents a significant limitation. To address this, we have revised the manuscript and added a clear statement acknowledging the restricted generalizability of our findings.

The Results section still includes two almost identical paragraphs both describing Table 1. Please, merge them into a single, short description.

Response: Thank you for the observation. For better visibility and to emphasize the results, we considered it necessary to present the findings both in the table and in written form, even if this implied some repetition. Thank you for your understanding.

Language and style still need some revisions: some grammatical errors, repeated expressions, and long sentences are still a bit impeding. A final language revision is required to improve clarity and scientific readability.

               Thank you for the observation. I have taken this aspect into consideration and have made the            necessary modifications. Please see the revised version of the manuscript.

Reviewer 2 Report

Comments and Suggestions for Authors

Dear authors,

All of the suggestions were addressed and the manuscript is suitable for publication.

Kind regards

Author Response

Dear reviewer

Thank you for your recommendations and for accepting the article for publication.

Reviewer 3 Report

Comments and Suggestions for Authors

Thank you for the revision. The manuscript has improved significantly. However, some minor issues remain in the manuscript.

The objective is a section of introduction rather than methodology.

Sample size calculation and  its rationale have not been addressed

Figures 1 and 2: image quality is poor

Author Response

Dear Reviewer,

               Thank you for your observations. I have added the modifications and explanations based on your recommendations. Please read the revised version.

The objective is a section of introduction rather than methodology. Sample size calculation and  its rationale have not been addressed. Figures 1 and 2: image quality is poor

Response to Reviewer:

Thank you for pointing out this important aspect. We have now addressed the issue directly in the revised manuscript. Specifically, we added a justification for the sample size in the  Discussion sections. We clarified that our sample size reflects the total number of eligible patients meeting strict inclusion criteria within the recruitment timeframe. Although the sample is limited, it remains consistent with exploratory cross-sectional studies in this field and ensures methodological rigor.